# Detection of Shelterbelt Density Change Using Historic APFO and NAIP Aerial Imagery

**Morgen W.V. Burke [1], Bradley C. Rundquist [2] and Haochi Zheng [1,*]**

[1] Department of Earth System Science and Policy, University of North Dakota, Grand Forks, ND 58202, USA; morgen.burke@und.edu
[2] Department of Geography and GISc, University of North Dakota, Grand Forks, ND 58202, USA; bradley.rundquist@und.edu
[*] Correspondence: hzheng@aero.und.edu; Tel.: +1-701-777-6056

**Abstract:** Grand Forks County, North Dakota, boasts the highest concentration of shelterbelts in the World. As trees age and reach their lifespan limits, renovations should have taken place with new trees being planted. However, in recent years, the rate of tree removal is thought to exceed the rate of replanting, which can result in a net loss of shelterbelts. Through manual digitization and geographic object-based image analysis (GEOBIA), we mapped shelterbelt densities in the Grand Forks County using historical and contemporary aerial photography, and estimated actual changes in density over 54 years. Our results showed a doubling in shelterbelt densities from 1962 to 2014, with an increase of 6402 $m^2/km^2$ over the 52 years (or 123 $m^2/km^2/year$). From 2014 to 2016, we measured 1,040,178 $m^2$ of shelterbelt areas removed from the county, creating a density loss of $-157$ $m^2/km^2/year$. The total change over two years was relatively small compared with that seen over the previous 52 years. However, the fact that the rate of shelterbelt planting has slowed, and more removal is occurring, should be of concern for an increased risk of wind erosion, similar to that experienced in Midwestern U.S. during the 1930s. The reduction of shelterbelt density is likely related to changes in farming practices and a decline in the Conservation Reserve Program, resulting from the increased returns of growing other row crops. To encourage shelterbelt planting as a conservation practice, additional guidelines and financial support should be considered to balance the tradeoff between soil erosion and agricultural intensification.

**Keywords:** geographic object-based image analysis; shelterbelts; Conservation Reserve Program

## 1. Introduction

Soil erosion of agricultural lands is a major contemporary global environmental problem. Arable soils provide important ecosystem services, with approximately 15 million $km^2$ used for crop production [1]. The demand for increases in farm land and crop yields is expected to rise with an estimated World population of 9.7 billion by the year 2050 [2]. Maintaining a healthy and productive soil is a critical issue for sustaining food production and other agricultural activities when meeting society's needs [1].

In the Midwestern U.S., the 1930s are remembered as a period of extreme soil loss when drought exposed top soil, resulting in high winds carrying huge volumes eastward, and depositing it into the Atlantic Ocean [3]. Dust storms experienced during the 1930s have been considered the worst human-driven environmental problem that the U.S. has faced [4]. Since then, changes in agricultural practices, such as the implementation of conservation tillage and the use of shelterbelts, have reduced the potential for wind erosion [5]. Nevertheless, recent estimates of wind erosion in U.S. cropland average 4.3 Mg/ha, while soil renewal rates are less than 1 Mg/ha. These data suggest that

policymakers and land managers should extend and strengthen the mitigation efforts to reduce wind erosion and preserve soil productivity [6].

Shelterbelts—sometimes-called windbreaks, hedgerows, or fencerows—have been used to mitigate wind erosion since the 1450s [7]. In the 1930s, the U.S. President Franklin Roosevelt established the Prairie States Forestry Project, with the goal of planting shelterbelts from North Dakota to Texas [7,8]. In addition to the reduced rates of soil erosion by wind, shelterbelts can maintain soil moisture which benefits crop yields, even outweighing the loss of acreages used for planting the shelterbelt [9]. Shelterbelts also improve crop water usage during drought periods by reducing evaporation rates; they reduce wind-chill impacts on livestock during winter, and improve livestock health; and reduce stress and damages on people and properties by providing protection from high winds [10]. Shelterbelts maintenance is important for preserving healthy trees, otherwise landowners must decide to renovate or remove tree's during their later life-stages [11]. In the U.S., many shelterbelts have been planted since the 1930s [8], species composition and tree density have experienced significant changes over the years. Baltensperger [12] measured the linear distances of shelterbelts in Iowa and Kansas and found that, in both states between the 1880s and the 1970s, shelterbelts decreased from 1600 km to 72 km in Iowa, and from 3000 km to 1100 km in Kansas. Schaefer, Dronen, and Erickson [11] surveyed 2875 shelterbelts in South Dakota and found that only 1150 were in a healthy condition, having no need for renovation.

Grand Forks County, North Dakota, once had a very high concentration of shelterbelts, and planting tree shelterbelts was a popular practice among the local communities to protect the soil (Figure 1) [13]. However, this point of view may have changed, recent interviews with land managers suggest that that the rate of shelterbelt removal in this region has increased [13–18]. Many shelterbelts have reached their life expectancy, and are in need of maintenance [15]. Agricultural producers are choosing to remove the trees without replanting, to alleviate the burden of maintenance costs [13,15]. This raises concerns that wind erosion could increase, especially during dry climate conditions. Soil erosion can occur at wind speeds above 6.2 m/s, which have been recorded in the county [19]. Understanding how shelterbelts have changed, and what may have driven these changes, in this region could provide scientific evidence and determine if extra policy measures are needed to cope with such changes.

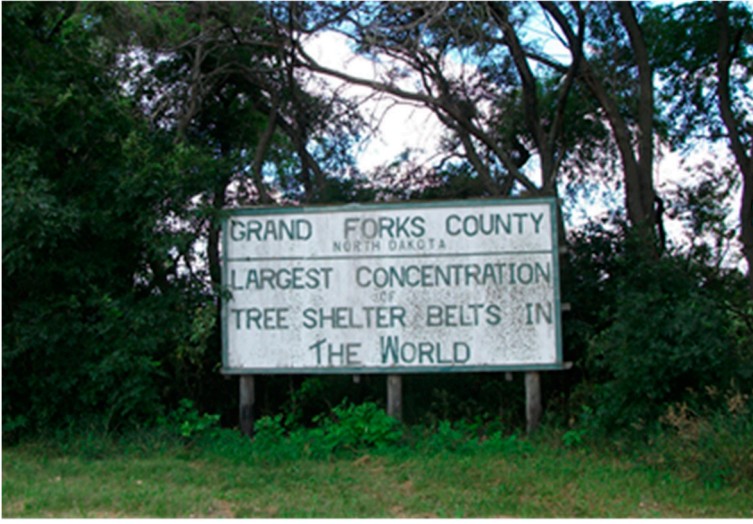

**Figure 1.** A roadside sign near Larimore, ND.

To investigate this spatial shift, we adopted remote sensing techniques for data acquisition, and used Geographic Information Systems (GIS) to analyze spatial data. Shelterbelts can be digitized using aerial imagery over multiple years, and changes in shelterbelt densities can be measured [20]. While several studies have used either aerial or satellite imagery to classify shelterbelts, e.g., [21–26],

none have attempted to detect change in shelterbelt densities over time. The ability to detect changing shelterbelt densities can provide insight into changes in landowner decision-making and support future policymaking on mitigating wind erosion.

## 2. Materials and Methods

### 2.1. Study Area

Grand Forks County (GFC; Figure 2) is located within the Red River Valley of North Dakota. The Red River of the North runs south to north along the eastern boundary of the county. GFC has a population of 70,795 [27], and comprises 372,021 ha of land, with 970 farms occupying 330,417 ha [28]. After retreat of the Glacial Lake Agassiz, the region was left with fertile, fine, and loamy soils that were well-suited for agriculture. The paleolake also left behind a landscape that had little topographic relief [29]. GFC is within the Great Plains region of the contiguous U.S., where 97% of land is non-forest, and instead consists mostly of agricultural land use and grassland vegetation [30]. In GFC, corn, soybeans, and spring wheat were the most common crops in 2017 with 16.5 million bushels of corn, 8.7 million bushels of soybeans, and 8.0 million bushels of spring wheat produced [31].

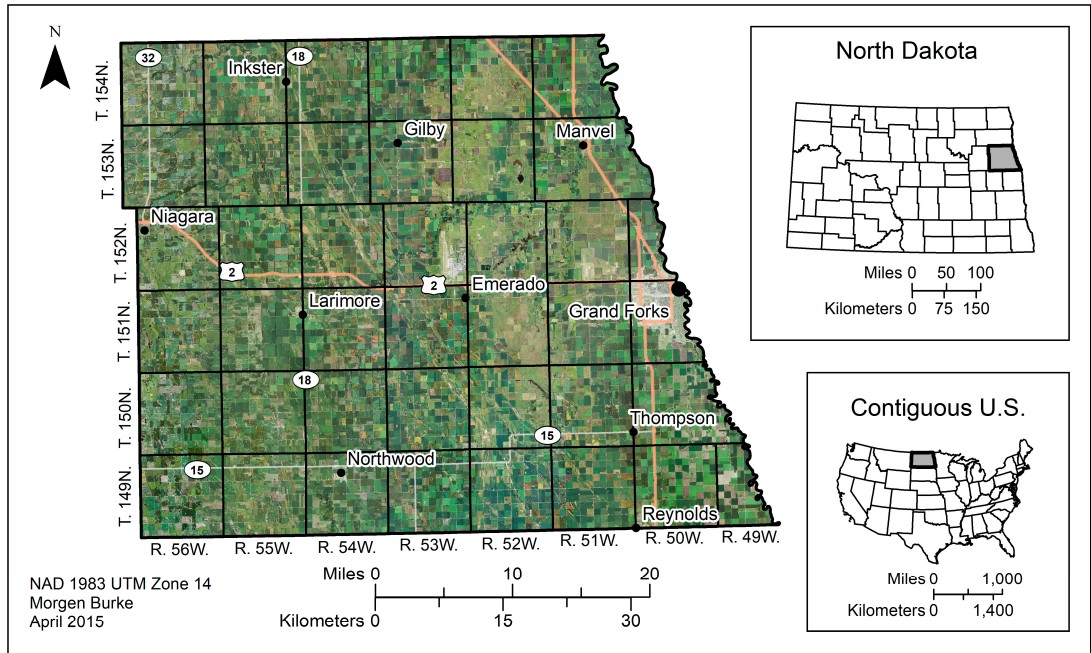

**Figure 2.** Grand Forks County, ND, showing townships and ranges from the Public Land Survey System (PLSS) with 2014 NAIP Aerial Imagery.

Shelterbelts are planted as long linear arrays at the edge of agricultural fields. They are often only 1–3 trees wide, and are typically between one half mile and a mile long (0.8–1.6 km) [10]. A survey of Montana and North Dakota shelterbelts recorded eight species of coniferous trees, with Colorado blue spruce (*Picea pungens*) and ponderosa pine (*Pinus ponderosa*) as the two most reported; 17 species of deciduous trees with green ash (*Fraxinus pennsylvanica*) and Russian-olive (*Elaeagnus angustifolia*) as the two most reported; and 13 species of shrubs with caragana (*Caragan arborescens*) and common lilac (*Syringa vulgaris*) as the two most reported [32]. Since alkaline soils are common in the county [33] tolerant tree species such as Green ash, Russian-olive, Arnold Hawthorn (*Crataegus arnoldiana*), and Siberian elm (*Ulmus pumila*) are available for planting [34].

## 2.2. Digitizing Historic Panchromatic Imagery

As a historic reference for shelterbelt densities in GFC, we acquired 1962 aerial imagery from the U.S. Department of Agriculture (USDA), Farm Service Agency (FSA), Aerial Photography Field Office (APFO). The 1962 imagery was the earliest available for GFC, and should capture the planting that took place after the Prairie States Forestry Project started in the 1930s. We received 832 digitally scanned panchromatic images taken at a scale of 1:20,000, and scanned to provide a spatial resolution of $25 \times 25$ cm. We first modified the imagery to remove all fiducial marks, borders, scanning errors, and index numbers. We then used ArcGIS Desktop 10.3 (Environmental Systems Research Institute, Redlands. CA) to georeference each image. We used the 2014 USDA National Agriculture Imagery Program (NAIP) imagery for GFC to align the 1962 imagery using the best available tie points for each image. The total root mean square error (RMS) for each image was less than or equal to 1 m, and at least 10 tie points were used for every image. Once each image had been georeferenced, they were mosaicked into a single 1962 image for GFC.

We manually digitized shelterbelts from the 1962 imagery using ArcGIS Desktop 10.3, to create a polygon around its perimeter for each visible shelterbelt. The 1962 imagery was kept at a fixed scale of 1:3000, and each section of land was checked one at a time for shelterbelts. The shelterbelts were drawn by having the image interpreter follow directly around the tree crowns. Trees were classified as a shelterbelt if they were adjacent to fields, which excluded those around buildings or farmyards. This helped to remove some variability from the data, by ensuring any detected change in shelterbelts was not due to changes in the number of farmyards, but instead because of cropland management decisions.

## 2.3. Geographic Object-Based Image Analysis of National Agriculture Imagery Program Imagery

Geographic object-based image analysis (GEOBIA) is a modern form of imagery analysis and classification that automates the extraction of image features by grouping image pixels into objects, and then classifying the objects based on spatial, spectral, and relational properties [35]. We used both the 2014 and 2016 NAIP imagery for GFC. These were the most recently acquired images that were captured during the summer time for the entire county, ensuring the trees still had green leaves. This was not always the case with NAIP imagery, for example the available imagery in 2015 was captured in late September when senescence had already set in. This imagery include red, green, blue, and near-infrared spectral bands, and have a $1 \times 1$ m spatial resolution. The 1962 imagery used in Section 2.2 had a higher spatial resolution, but was not resampled since it was not used in the GEOBIA process, and resampling would only hinder image interpretation.

We executed the GEOBIA using the eCognition Developer 9.1.2 software (Trimble Geospatial, Sunnyvale, CA, USA). We produced two additional data layers to the NAIP imagery to improve classification: a normalized difference vegetation index (NDVI; Equation 1), and a green normalized difference vegetation index (GNDVI; Equation 2) image for both 2014 and 2016. Once we loaded the NAIP imagery into eCognition, the imagery was segmented using the multiresolution segmentation toolset with a scale of 10, a shape of 0.2, and a compactness of 0.9 (Figure 3). These values were taken from Meneguzzo, Liknes, and Nelson [24], except for scale which was changed from 15 to 10 after trial-and-error showed it produced more desirable object sizes. In the multiresolution segmentation, we weighted the four spectral bands so that near-infrared had a weighting of three while red, green, and blue all had a weighing of one, this improved segmentation of vegetation objects from other features in the imagery. Using the means of the two vegetation indices, we extracted vegetation objects from the segmented image objects. Following the framework established by Meneguzzo, Liknes, and Nelson [24] as a starting point for detection of tree objects and then modifying it according to Wiseman, Kort, and Walker [21], we were able to classify shelterbelt objects. To classify tree image objects we used several object features including: NDVI and GNDVI mean values, object texture, length-to-width ratio, object symmetry, standard deviation, mean object value, and mean difference between adjacent objects. By using image object size, shape, compactness,

and length-width parameters, we separated shelterbelts from the general tree object class. Relational information allowed us to further improve classification, such as identifying trees next to objects that were likely tree shadows. We ran this process twice, producing two sets of shelterbelt polygons, one for 2014 and one for 2016. While similar image-object properties where used for extracting shelterbelts from the two NAIP images, many of the threshold values were adjusted to account for variation in pixel values between individual images. As noted by Wiseman, Kort, and Walker [21], determining the proper spectral, spatial, and relational values for feature extraction is very much a trial-and-error process that requires adjusting, testing, visually assessing, and repeating the feature extraction process several times to find an acceptable outcome.

$$\text{NDVI} = \frac{(NIR - \text{Red})}{(\text{NIR} + \text{Red})} \tag{1}$$

$$\text{GNDVI} = \frac{(NIR - \text{Green})}{(\text{NIR} + \text{Green})} \tag{2}$$

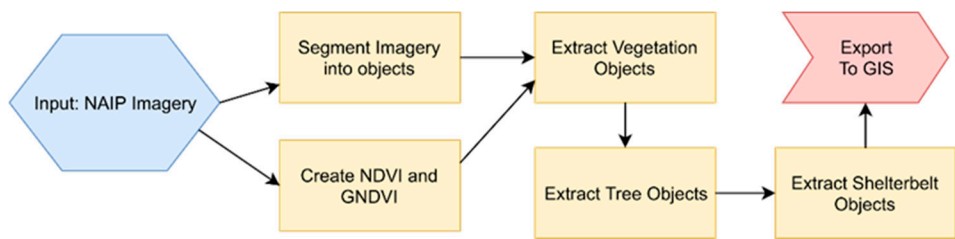

**Figure 3.** Geographic Object-Based Image Analysis workflow model to extract shelterbelts from National Agriculture Imagery Program imagery.

To ensure the quality of the shelterbelt polygons, the GEOBIA-produced shelterbelts were manually validated to remove inaccuracies from the GEOBIA classification. Similar to the 1962 shelterbelts, trees were classified as a shelterbelt if they were a linear feature planted adjacent to fields, and excluded if they were around buildings or farmyards. A fixed scale of 1:5000 was used, and each section was visually checked one at a time to remove any polygons that were improperly classified as shelterbelts, or add any shelterbelt polygons that were missed. The manually digitized shelterbelts were created with a slightly smaller scale of 1:3000 for further reduction of time and effort in checking the GEOBIA produced shelterbelts.

*2.4. Geographic-Object-Based Image Analysis for Change Detection*

Once the shelterbelts were extracted from the three years of data (1962, 2014, and 2016), they were compared against one another to detect changes. However, over time, the width of many individual shelterbelt canopies changed, resulting in a difference in the shape of the extracted shelterbelt polygons, which would produce a false positive for change. This problem was noted by Burke [20], and resulted in a 35.8% increase in detected change, mostly because of canopy growth between 1962 and 2014. To resolve this issue, we compared shelterbelts against one another across the three years, and any overlapping polygons were detected and modified into a single identical polygon representing the shelterbelt footprint for that location.

eCognition Developer 9.1.2 was used to conduct the change detection across the three years. Three object data layers were created for each year, and then overlaps were detected across the three layers. In total, there are seven possible outcomes when overlapping polygons among the three layers. Polygon can have no overlap with any others (x3); or overlap with one of the others (x3); or overlap across all three layers (x1) (Figure 4). We used a search radius of 30 m, so that any polygons that were found within the radius of one another would be considered overlapping polygons. To minimize the potential effect of an increased canopy spread and avoid overestimating the actual area, we chose the

smallest overlapped-polygons, and placed it into the appropriate object layer. Once the overlapping polygons were processed, the final object layers were aggregated to produce three final polygon layers, one for each year, that were brought into a GIS for further analysis.

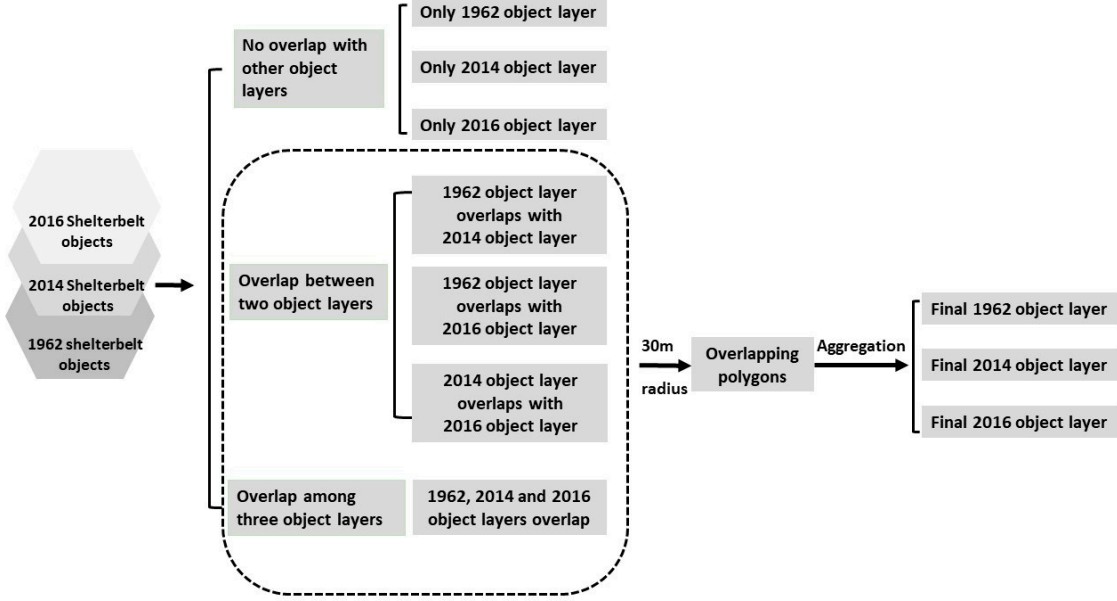

**Figure 4.** Flowchart showing the possible outcomes from change detection across three object layers.

*2.5. Change Detection Accuracy Assessment*

A randomized accuracy assessment of shelterbelt change detection was carried out using 500 points or locations randomly selected in areas of detected change. The points would only be placed in a location that was detected as being a shelterbelt in either one or two of the three years. In other words, any point found to be located on a shelterbelt for all three years, or not on a shelterbelt for all three years, resulted in an error in change detection. This was used to determine if the manually digitized and GEOBIA produced shelterbelt polygons could be compared within an acceptable level of accuracy. We adopted an accuracy between 92–96% based on other GEOBIA studies [21,23,24,26].

*2.6. Examining Shelterbelt Density Change and Spatial Patterns in GFC*

To measure shelterbelt density, we aggregated polygons by unit area. While on-the-ground studies often report shelterbelt density as a measure of length (m) over unit area [11,12], we chose shelterbelt area ($m^2$) as an acceptable proxy of length based on the nature of image classification. In the U.S., the Public Land Survey System (PLSS) divides most land areas into section-township-range coordinates of approximately 259 ha (2.59 $km^2$) [36]. We measured shelterbelt density per (PLSS) section, and used these density measurements to carry out our assessment of density change (Equation (3)).

We also examined the spatial relationship between shelterbelts in GFC and a few independent factors. We assessed several geographic, topographic, and social-economic factors including soil potential productivity, land capability, slope, wind erodibility, assessed land value, soil pH, surface geology, and shelterbelt ownership. Climate information such as precipitation and temperature were not incorporated in our study because the recorded climate information at local weather stations is limited and would not be able to explain detailed spatial variations over the entire county. Furthermore, we only measured the density for three points in time, which would not be sufficient for conducting statistical analysis. However, we do recognize that climate likely plays a role in the use of shelterbelts.

For instance, the drought conditions seen in the 1930s were the main reason the Prairie States Forestry Project was initiated and resulted in planting many shelterbelts.

$$\text{Shelterbelt Density} \left( m^2/\text{k}m^2 \right) = \frac{\Sigma \; \text{Shelterbelt Area} \left( m^2 \right)}{\text{PLSS Section Area} \left( \text{k}m^2 \right)} \qquad (3)$$

We acquired information on soil pH from the USDA Natural Resource Conservation Service (NRCS) Soil Survey Geographic Database (SSURGO) to compare against the shelterbelt density data. We chose to examine soil pH because GFC is known to have alkaline soils, which can effect plant communities [33]. Shelterbelt establishment can be hindered in saline and alkaline soils unless a tolerant tree species, such as green ash or Russian-olive, is planted [37,38]. We expected to see lower shelterbelt densities in parts of the county where alkaline soils could hinder tree establishment. The SSURGO soil pH value represents the negative logarithm to the base 10, of the hydrogen ion activity in the soil using a 1:1 soil–water ratio. To calculate soil pH for each section, we assigned the value from a SSURGO polygon if it held the majority of a PLSS sections area. Then we used the software GEODA (University of Illinois, Champaign-Urbana, IL) to perform a Bivariate Local Moran's I, to test for spatial autocorrelation between soil pH and shelterbelt density at a 95% confidence interval [39]. We ran the Bivariate Local Moran's I using queen contiguity over 999 permutations.

We also compared shelterbelt density with GFC surface geology. We acquired GIS ready surface geology data from the North Dakota Geological Survey (NDGS), providing the distribution of distinct geological units in the county. To compare this information with our shelterbelt density data we assigned each PLSS section with a surface geology type based on the surface geology that contained the majority of a section's area. At the minimum 13 sections were classified as sand, while at the maximum 702 sections were classified as till. Then a one-way analysis of variance (ANOVA) was performed to determine significance in shelterbelt densities between geological units, sand, silt, till, clay, and cross-bedded sand (Equation 4). The shale and calcareous shale surface geology categories were excluded from the ANOVA since their total area combined made up less than 0.01% of the area in GFC. We ran three ANOVA tests, one for each year of shelterbelt data, against the individual surface geology layer to see if significant differences would remain consistent. The ANOVA was conducted with a 95% confidence interval, followed by a robust comparison of means that allowed for unbalanced group sizes, non-normality, and heteroscedasticity. This was completed using the statistical software package R 3.5.0 (R Foundation for Statistical Computing, Vienna, Austria), using the methods described by Herberich, Sikorski, and Hothorn [40].

$$\begin{array}{c} H_0 : \mu_1 = \mu_2 = \mu_3 \ldots = \mu_j \\ H_a : \text{Mean shelterbelt densities for each surface geology category are not equal} \end{array} \qquad (4)$$

Lastly, we used Common Land Unit (CLU) to assess the distribution of shelterbelts based on the boundary of agricultural management [41]. We assumed that if land managers and farm operators' valued and had equal access to plant shelterbelts, with everything else being constant to disturb landowner's decision-making, then the percentage of shelterbelts within each CLU should follow a normal distribution. We measured the total area of shelterbelts per each CLU, and then divided it by CLU area to derive the shelterbelt coverage of each CLU. We examined owner and operator distributions for normality using a histogram.

## 3. Results

### 3.1. Shelterbelt Density

In 1962, GFC had 22,352,136 m$^2$ of shelterbelts; this increased to 43,505,202 m$^2$ in 2014, and fell to 42,465,024 m$^2$ in 2016. In the latest USDA Census of Agriculture, GFC had 3304 km$^2$ (816,478 ac) of farmland [28], which resulted in shelterbelt densities of 6765 m$^2$/km$^2$, 13,167 m$^2$/km$^2$,

and 12,853 m$^2$/km$^2$ for 1962, 2014, and 2016, respectively. From 1962, the density had more than doubled by 2014, and decreased from 2014 to 2016.

Figure 5 demonstrates the spatial pattern of shelterbelt density, which was not uniform across the county. A band with the highest densities (>9538 m$^2$/km$^2$) runs north-to-south through the center of the county, with sections of low density on either side. In 1962, 50% of the sections had no shelterbelt; by 2014, this had decreased to 29%, and then increased to 31% in 2016. This diffusion of shelterbelts throughout the county is visible in the 1962 to 2016 density change maps in Figure 5. From 1962 to 2014, 55% of sections showed a positive increase in density, while 18% were negative; for 1962 to 2016, these values were 53% and 19%, respectively. From 2014 to 2016, only 12% of sections showed a positive increase in density while 30% were negative.

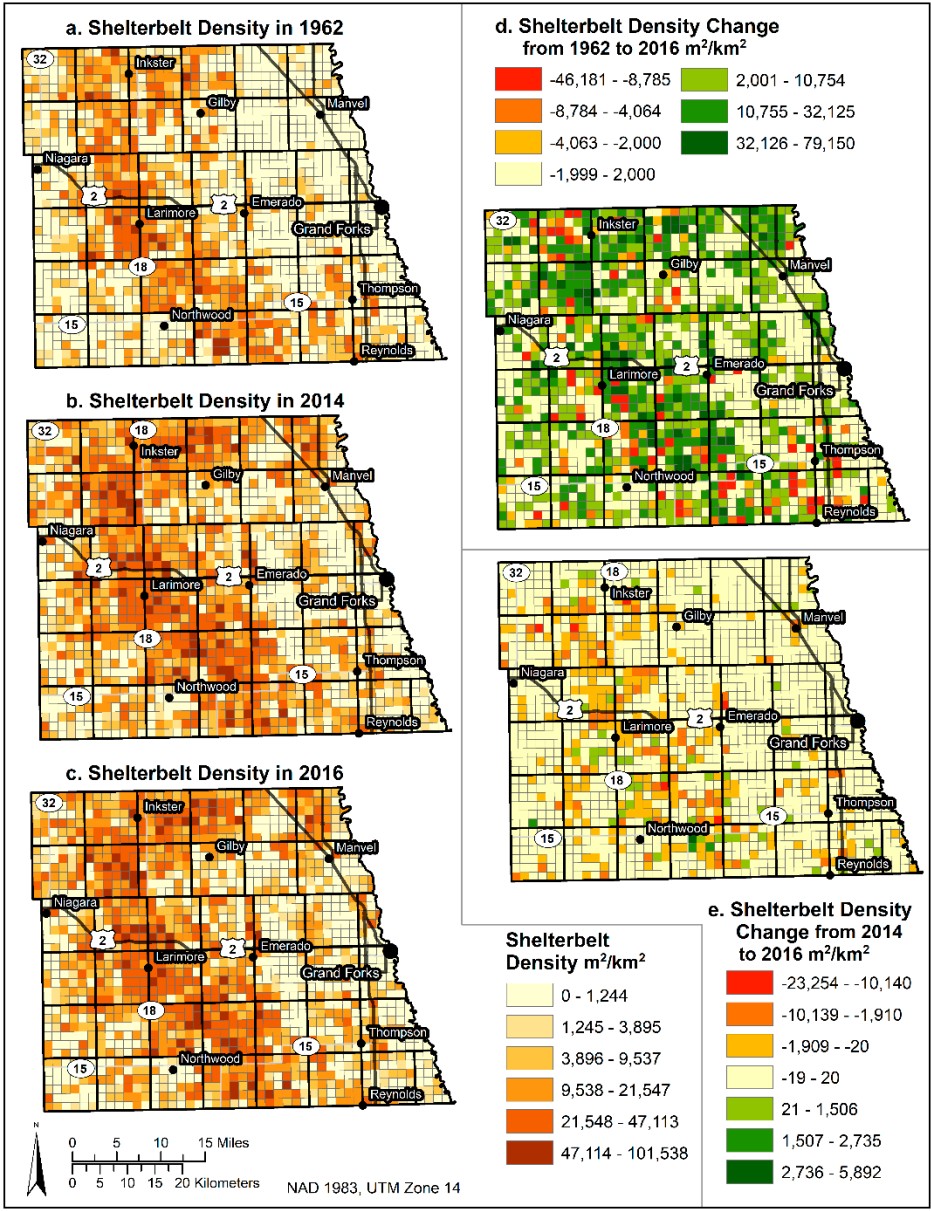

**Figure 5.** Grand Forks County shelterbelt density measured per section of land for the three years of data: (**a**) 1962, (**b**) 2014, and (**c**) 2016, as well as the change in shelterbelt density (**d**) 1962–2016, and (**e**) 2014–2016.

Changes in shelterbelt density from 1962 to 2016 was not uniform across GFC, mostly occurring with a band running north to south, which was consistent with the spatial pattern of existing

shelterbelts in GFC. Sections of land containing a shelterbelt went from 50% to 71% between 1962 and 2014. The newly added sections saw the most growth in density with a mean of 10,593 $m^2/km^2$, while sections of land that already had a shelterbelt increased by an average of 7144 $m^2/km^2$. This demonstrates that the growth of shelterbelts mainly occurred in the extensive margin, i.e., expansion into sections that previously did not contain a shelterbelt, rather than into the intensive margin, thus intensification in sections with existing shelterbelts. By separating sections based on whether density increased or decreased, we found that from 1962 to 2014, 18% experienced a decline in shelterbelt density averaging $-5493$ $m^2/km^2$, while growth in density occurred in 55% of sections averaging 12,199 $m^2/km^2$.

To validate the GEOBIA produced shelterbelts polygons compared to the manually digitized shelterbelts polygons, we randomly selected 500 points or locations in areas of detected change (Table 1) for an accuracy assessment. Each point was visually interpreted as either being a shelterbelt (Yes) or not a shelterbelt (No) for the three years of imagery. In total, only 22 of the points (4.4%) failed the accuracy assessment, resulting in an overall accuracy of 95.6%. Of these, 15 (3.0%) were inaccuracies in detecting shelterbelt removal, while 7 (1.4%) were inaccuracies in detecting shelterbelt additions.

**Table 1.** Change detection accuracy assessment. This error matrix was used to assess the overall quality of detected change between the 1962, the 2014, and the 2016 digitized shelterbelt polygons. The assessment used 500 points placed randomly in areas of detected change to test if change occurred or not. An inaccurately detected change resulted from points that either had a shelterbelt in all three years 1962, 2014, and 2016, or not (Yes, Yes, Yes or No, No, No). All other possible combinations resulted in an accurately detected change.

| Was a Shelterbelt in: | | | Count out of 500: | |
|---|---|---|---|---|
| 1962 | 2014 | 2016 | Accurate | Error |
| Yes | Yes | Yes | NA | 15 |
| Yes | Yes | No | 2 | NA |
| Yes | No | Yes | 0 | NA |
| Yes | No | No | 64 | NA |
| No | Yes | Yes | 404 | NA |
| No | Yes | No | 8 | NA |
| No | No | Yes | 0 | NA |
| No | No | No | NA | 7 |
| | | Sum | 478/500 | 22/500 |
| | | % | 95.6% | 4.4% |

The location of shelterbelt change between 2014 and 2016 was not different from the previous change occurring in the region containing the majority of shelterbelts. However, the direction of density change appeared to have shifted from positive to negative, with 30% of sections experiencing a decline in density averaging $-1143$ $m^2/km^2$, while only 12% of sections experienced growth averaging 191 $m^2/km^2$. While the total change over two years was relatively small compared with that seen over the previous 52 years, the rate of growth was less than half going from 235 $m^2/km^2/year$ to only 96 $m^2/km^2/year$, and the rate of decline increased five-fold from $-106$ $m^2/km^2/year$ to $-572$ $m^2/km^2/year$.

*3.2. Shelterbelt Density and Soil Alkalinity*

We ran a Bivariate Local Moran's I to test the spatial autocorrelation between shelterbelt density and soil alkalinity based on the SSURGO soil pH data within GFC (Figure 6). Soil pH in the county ranged from 6.5 (close to neutral), to more alkaline soils with a pH of 8.2. The Bivariate Local Moran's I identified the significantly high shelterbelt densities in the central portion of the county, and significantly low densities along the eastern side of the county. This showed an inverse correlation with soil pH, where many portions of the eastern side of the county had soils that were more alkaline, while the central portion of the county was more pH neutral.

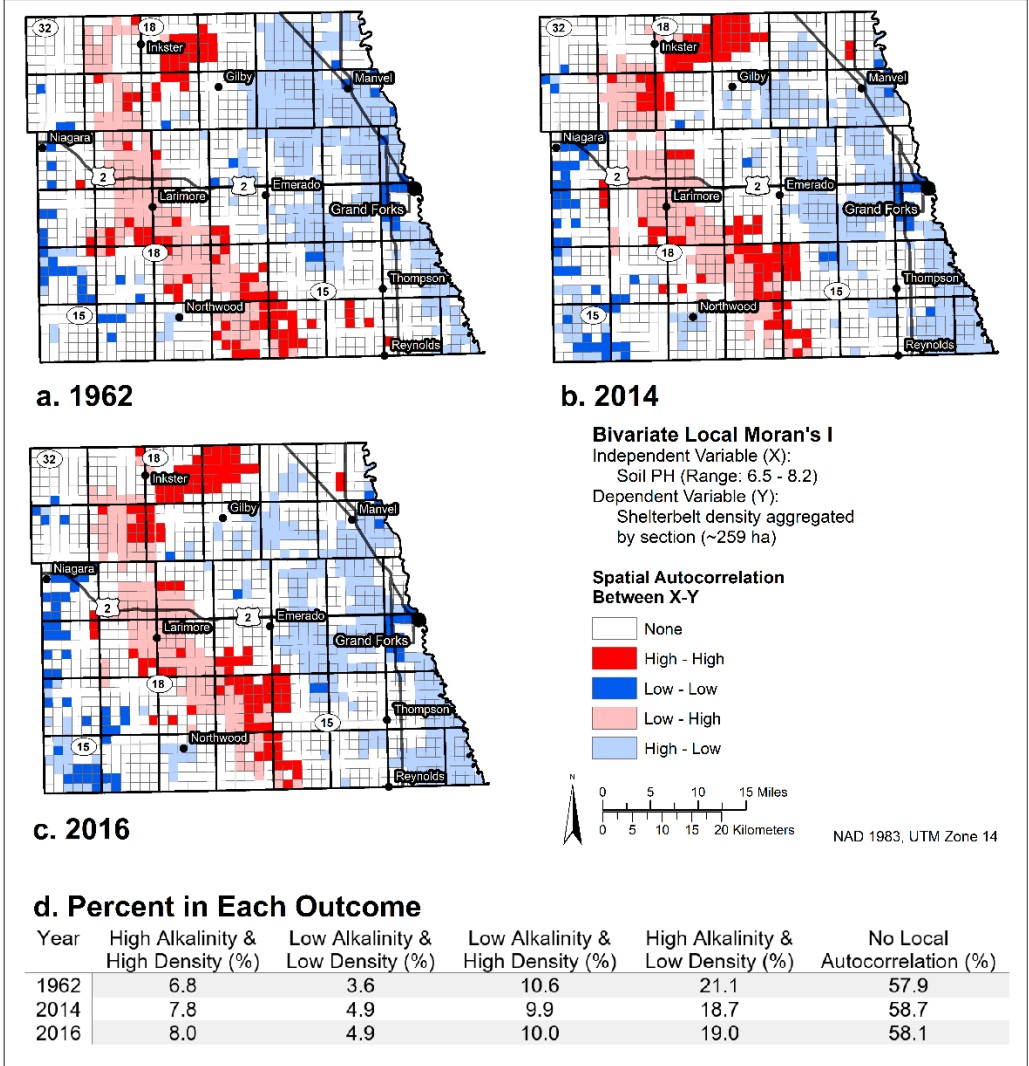

**Figure 6.** Bivariate Local Moran's I testing the spatial autocorrelation between shelterbelt density and the Soil Survey Geographic Database soil pH, set at a 95% confidence interval for (**a**) 1962, (**b**) 2014, and (**c**) 2016. (**d**) The percentage of sections that fell under each possible outcome.

*3.3. Shelterbelt Density and Surface Geology*

We compared the three years of shelterbelt densities in GFC against the individual surface geology layer using three one-way ANOVA tests [40], set at a 95% confidence interval (Figure 7). The three years of shelterbelt data were used to check the robustness of the statistical significance. There were 1390 sections of land used to calculate shelterbelt densities for the one-way ANOVA, of these sections; 702 were till, 398 were silt, 238 were clay, 39 were cross-bedded sand, and 13 were sand. Significant differences between surface geology categories were found across all three years with $p < 0.001$. In general shelterbelts on sand had a significantly higher mean density than clay, silt, and till, except for clay in 1962, ranging from 21,699 $m^2/km^2$ in 1962 to 36,763 $m^2/km^2$ in 2014. Shelterbelts planted on silt had a significantly low mean density across all three years ranging from 2613 $m^2/km^2$ in 1962 to 7456 $m^2/km^2$ in 2014. Densities within areas of clay and till fell somewhere between those of sand and silt across all years, and were only significantly different from each other in 1962.

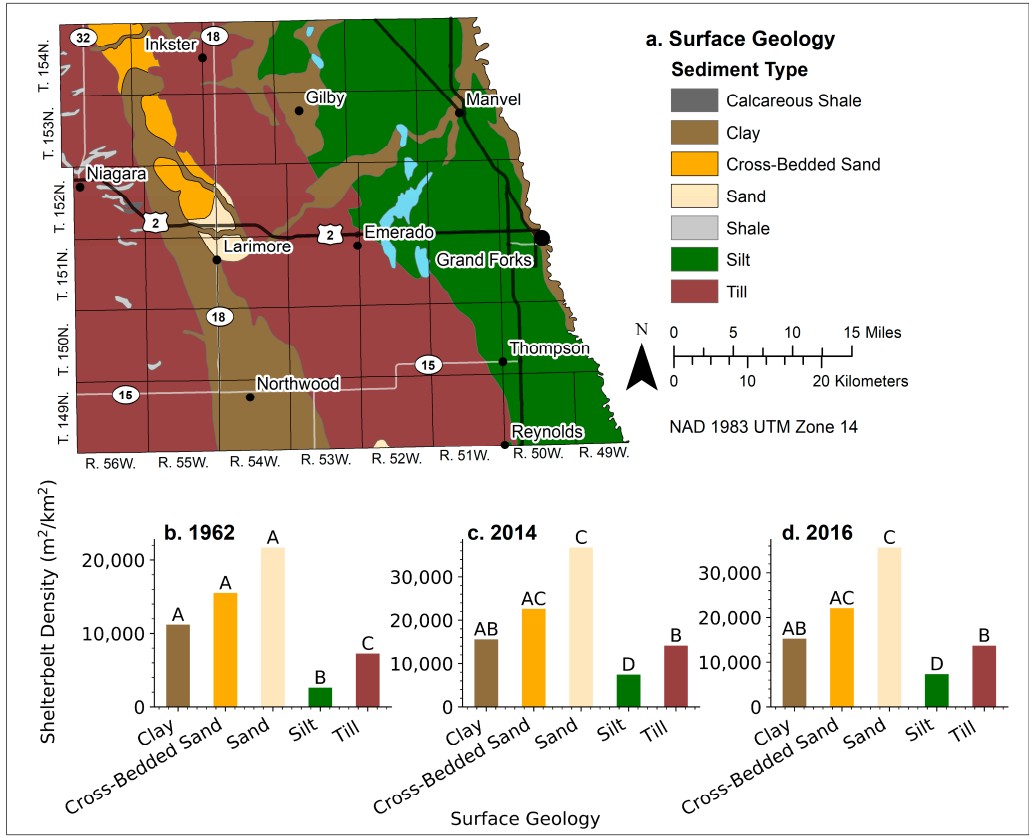

**Figure 7.** (**a**) Spatial arrangement of surface geology found in GFC. The mean shelterbelt density for the five prominent types of surface geology within GFC, using density values from the three years: (**b**) 1962, (**c**) 2014, and (**d**) 2016. The lettering placed above the bars depicts significance levels from the one-way analysis of variance at a 95% confidence interval, representing a significant difference in shelterbelt density when compared with the other types of surface geology.

### 3.4. Shelterbelt Ownership

We examined shelterbelt ownership in GFC by measuring shelterbelt densities for each CLU based on the boundary of land management and farm operation using the 2014 data only. In both cases, shelterbelt ownership did not follow a normal distribution and was instead heavily skewed towards a few operations having the highest shelterbelt densities (Figure 8). For the 1750 units, 40.5% did not have a shelterbelt, while 17.8% had a shelterbelt with less than 7500 $m^2/km^2$, these values were 55.0% and 11.9% for the 5608 farm operators, respectively. Only 14.9% of units and 15.1% of operators had a density above 25,000 $m^2/km^2$, and among them only 1.3% of units had a density value above 100,000 $m^2/km^2$. The mean shelterbelt density of land units was 12,016 $m^2/km^2$, while the median was 2464 $m^2/km^2$; for operators mean density was 11,522 $m^2/km^2$, and the median was 0.0 $m^2/km^2$.

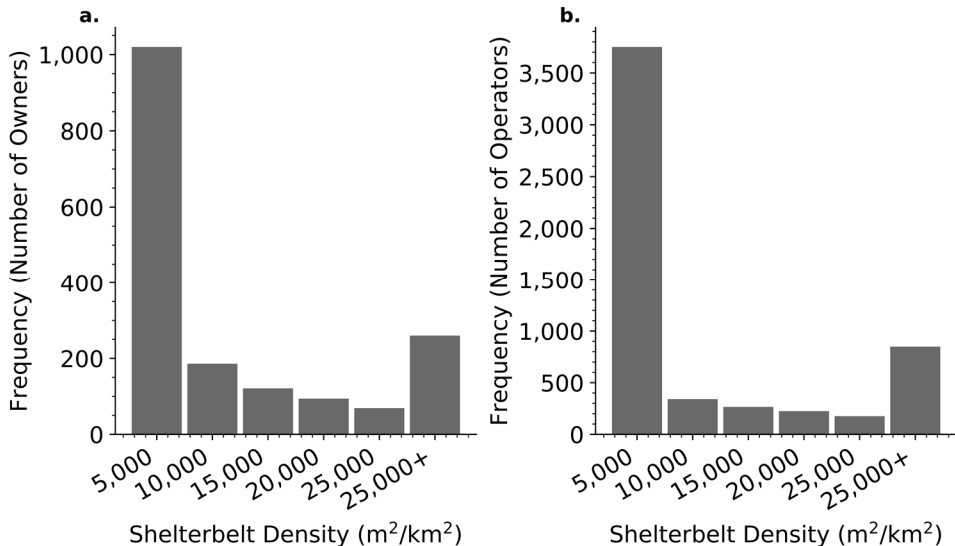

**Figure 8.** Distribution of total shelterbelt density in GFC owned by (**a**) all landowners or (**b**) farm operators in 2014.

## 4. Discussion

Using manual digitization and geographic object-based image analysis, we measured shelterbelt densities in GFC with multiple years of high-resolution aerial imagery. The result of density change between years was validated with an error of 4.4%. This amount of error is considered acceptable when compared with 4–8% errors produced by other studies for a single year classification [21,23,24,26].

Our results show a doubling of shelterbelt densities from 1962 to 2014, with an increase of 6402 m$^2$/km$^2$ over the 52 years (123 m$^2$/km$^2$/year; Figure 5). This large growth in shelterbelt density was surprising since reports before 2014 found that shelterbelt removal was becoming common practice [13–15]. These data suggest that shelterbelt density was much higher prior to our 2014 measurement and has started declining recently. Although the change is subjected to measurement errors, from 2014 to 2016, we detected 1,040,178 m$^2$ (2.4%) of shelterbelt area removed from the county, creating a density loss of $-157$ m$^2$/km$^2$/year.

Depending on individual species and their growth, it is possible that between 2014 and 2016 some tree planting was not detectable in the NAIP imagery, since small tree sapling may not be visible from remotely sensed imagery. However, other studies have documented that the number of tree planting since 2010 has declined to less than 20% the annual amounts planted in the 1990s [20], which has resulted in a 70% drop in tree sales in North Dakota between 2002 and 2013 [16,42]. Therefore, a more reasonable explanation for the lower density is that less planting has occurred.

The use of shelterbelt areas as a measurement for examining shelterbelt change has a few challenges. In Section 2.4 we addressed one of these challenges by removing changes in shelterbelt areas caused by increases or decreases in the tree canopy. Some studies have used linear lengths of shelterbelts as a metric for measurement [11,12], however these measurements were done in the field, or manually digitized with a much smaller sample. Using GEOBIA allows for the automation of shelterbelt feature extraction from imagery over large areas. The produced GEOBIA shelterbelts are polygons instead of lines and are measured in terms of area [21–26]. We further conducted a statistical analysis and validated that shelterbelt area (m$^2$) can serve as a close proxy of shelterbelt length (m) for measuring temporal changes (Figure A1). Our results also show that average shelterbelt length was consistent across all three years ranging from 436.2 m to 450.9 m (Table A1).

An interesting discovery from our shelterbelt classification was the spatial clustering of shelterbelts within GFC. Since most of the county is agricultural land, we assumed that shelterbelts would be evenly distributed throughout the region. Using soil pH with the Bivariate Local Moran's I (Figure 6) we found local regions of GFC that have significant spatial autocorrelation ($p \leq 0.05$) with shelterbelt

density. Much of the eastern side of the county, making up between 18.7–21.1% of sections, contains regions of high soil alkalinity and low shelterbelt density. These regions are likely not well suited for tree growth and shelterbelt establishment [38]. The majority of sections with a high shelterbelt density consisted in areas of low alkalinity, between 9.9–10.6% of sections, though regions of high density also exist in areas with alkaline soils, making up around 6.8–8.0% of sections. The results from analyzing soil pH data indicates that parts of eastern GFC would be more difficult for establishing shelterbelts or would require tolerant species.

Our analysis of surface geology (Figure 7) in GFC proved to have a consistent relationship with shelterbelt density. The regions of sand, cross-bedded sand, clay, and till in the county occur in a band running directly parallel with the locations of majority of shelterbelts. This is not a coincidence since this north-south band of clay and sandy soils are likely highly erodible soils made up of the beach ridges formed by Glacial Lake Agassiz [29]. Results from the ANOVA tests showed a significant relationship ($p \leq 0.05$) between the highest shelterbelt density and the sand surface geology. This relationship was consistent across all three years, with cross-bedded sand also falling into the highest shelterbelt density category for 1962 only. This suggests that agricultural operators likely plant shelterbelts in sandy soil due to its vulnerability to wind erosion [43]. The lowest shelterbelt densities across all three years of data ($p \leq 0.05$) occur in the silty soils found along the eastern side of GFC. These soils are likely more resistant to wind erosion than the sandy soils [43], and many of them are also alkaline soils which may not be suitable for shelterbelt establishment, particularly if alkalinity levels are high enough to hinder the growth of tolerant species.

Perhaps the most important factor in both understanding the placement of shelterbelts and their density dynamics is human-decision making, as Schaefer, Dronen, and Erickson [11] found in their examination of shelterbelts in Kansas. Shelterbelts that were well maintained, especially during their first few years of growth, were more likely to remain in a healthy condition longer, and those that were not cared for, were likely to need replacement or removal in later stages of life. We examined the ownership of shelterbelts in GFC (Figure 8), and found a heavy skew towards a few agricultural operators owning the highest densities of shelterbelts. In 2014, 55% of operators did not have a shelterbelt, while 40% of landowners did not have one. Of the remaining percentages that own a shelterbelt, the majority had a density less than 15,000 $m^2/km^2$, while only 1.3% of operators had densities above 100,000 $m^2/km^2$. This skewed distribution between both operators and landowners suggests that decision-making for the planting, maintenance, and removal of shelterbelts is likely influenced by personal preference and knowledge, as well as the physical condition of the soils owned.

In 1985 the Conservation Reserve Program (CRP) was initiated by the USDA FSA, which allowed land owners to voluntarily place land into conservation efforts over a 10–15 year period, and receive financial assistance through contract [44]. Under CRP, shelterbelts can fall into two categories: field windbreak (CP5) and shelterbelts (CP16), both serving the purpose of wind erosion control for crops, livestock, homesteads, and snow accumulation [45]. Data available from the USDA (Figure 9a) shows a steady increase in the hectares of shelterbelts enrolled in CRP contracts in North Dakota from the early 2000s until ~2010–2012. After 2012, shelterbelt enrollment declined across the state. This statewide summary aligns well with our detected shelterbelt change in GFC, showing an increase in shelterbelts up 2010, and then a sudden shift to a decrease. Recent changes in agricultural and energy policies to increase the use of biofuels created a greater demand for corn and soybeans in the Midwestern region of U.S. [46]. Following a major agricultural drought event in 2012, the prices of corn and soybeans have doubled in the region (Figure 9b). The surge in crop prices resulted in less interests in CRP enrollments, and in North Dakota an estimated 89,000 ha of grassland was converted to corn, and 220,000 ha of grassland was converted to soybeans [46]. The loss of CRP contracts during the high commodity prices in 2012 has been reflected by the change in the CRP shelterbelts (Figure 9a), suggesting that trees were removed to increase the amount of available land for corn and soybeans.

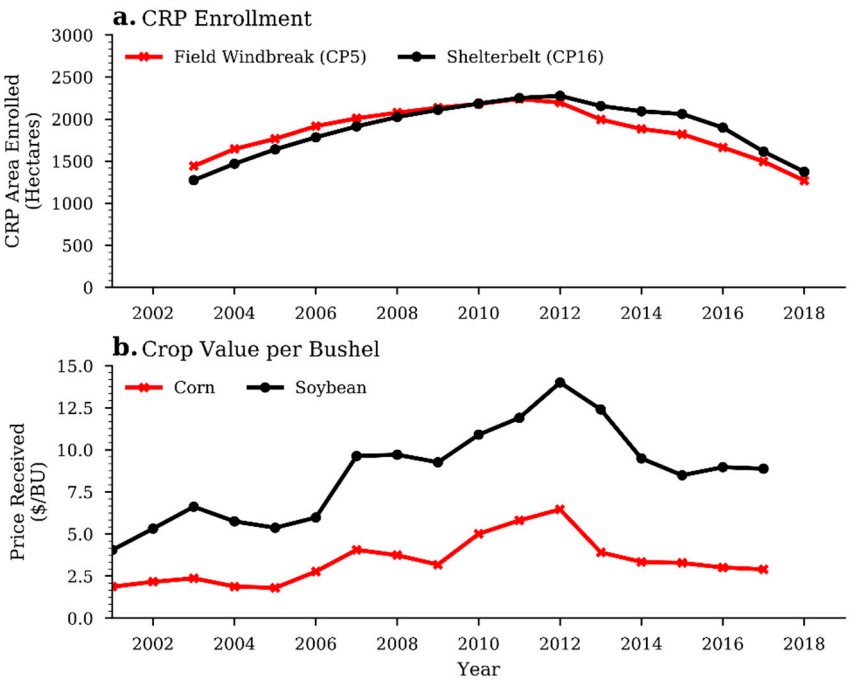

**Figure 9.** (**a**) The enrolment of shelterbelts in the conservation reserve program (CRP) in North Dakota (USDA FSA), and (**b**) the average annual price per bushel received for corn and soybeans in North Dakota (USDA National Agricultural Statistic Service (NASS)).

Congruently with the increase in crop prices, a shift in agricultural practice can also partially explain shelterbelt removal in GFC. In North Dakota, between 1982 and 2007, the midpoint acreage of farm sizes more than doubled [47]. In GFC, the average farm operation sizes increased from 32,784 ha in 1997 to 59,255 ha by 2012 [31]. This increase in farm size created the need for larger, more efficient machinery [47]. As agricultural machinery has gotten larger, it is more difficult to move in and out of fields, or to turn equipment at the edge of the fields when trees are in the way. Mature trees can get in the way of operation when branches or entire trees fall onto the field. In addition, herbicides in the U.S. were applied to 5–10% of cropland in the 1950s, and this increased to 90–99% by the 1980s [48]. During herbicide application, particularly with aerial sprays, herbicides could land on and damage the adjacent shelterbelts. The issues related to farming scale as well as management practices likely will influence landowner's decision-making related to shelterbelts' planting, maintenance, and removal [13–17], and ultimately has an effect on wind erosion in the future. On the other hand, outreach and information sharing with landowners and agricultural operators through extension services and government agencies could offer future opportunities in promoting shelterbelts planting as a conservation practices to mitigate soil erosion. In particular, financial support from government agencies can provide incentives to agricultural operators for shelterbelts planting, maintenance, and renovation. In 2015, the North Dakota Outdoor Heritage Fund awarded the North Dakota Forest Service $1.8 million to renovate shelterbelts in the state [18]. Continued monitoring of shelterbelts at a statewide level is needed to evaluate if conservation efforts like this can generate a substantial and long-term impact.

## 5. Conclusions

The aim of this study was to detect shelterbelt density change through time and determine if recent claims of tree removal should be of concern. We were able to use high-resolution aerial photography to both classify and detect change in shelterbelt densities within GFC. From 1962 to 2014—a total of 52 years—we found that shelterbelt densities doubled, and this provides great praise for the shelterbelt planting efforts that have taken place in the county. However, our more recent change investigation

from 2014 to 2016 suggests that the rate of shelterbelt planting in GFC has slowed, and more removal is taking place. While our change detection between 2014 and 2016 could have resulted from the 3% measured error in detecting shelterbelt removal (Table 1), the decline in CRP enrollment from 2012 to 2018 (Figure 9a) agrees with this trend. The decline in trees sales in North Dakota [42] also agrees with our finding that tree planting has declined. This raises concern for the potential risk of wind erosion in the future. If the increased rate of shelterbelt removal does not slow, or is not offset by an increase in planting, we could see increasing amounts of top soil loss during high wind events, particularly when the soil is exposed during crop planting and harvest.

The methods we developed for this study can be applied to measure shelterbelt densities in other parts of the Midwestern U.S. wherever NAIP imagery is available. Unfortunately, while the spectral, spatial, and relational object properties chosen for the 2014 and 2016 NAIP imagery can be used as a framework for automating the extraction of shelterbelt polygons from NAIP imagery (Figure 3), the exact values cannot be used directly since pixel values are not standardized. It would be beneficial to improve the GEOBIA process by developing a system that automatically adjusts threshold values for various imagery. This would simplify sharing and widespread use of the developed GEOBIA classification systems.

From a policy perspective, our findings indicate knowledge gaps in understanding changing agricultural practices related to shelterbelts. Our results show that individual landowner's and farm operator's preferences likely have a great influence on the number of shelterbelts planted in any particular field. To identify further the driving factors of decision-making on shelterbelts planting, a survey or other method of outreach will need to be incorporated. It is important to understand why agricultural producers that had shelterbelts in the past are no longer choosing to replace them. If changes in the farming scale contribute to the recent shelterbelt removal, policymakers should provide additional guidelines and incentives to balance the tradeoffs between soil erosion and agricultural intensification. Improvements in our understanding of changing agricultural technology will also be helpful to sustain the efforts in preserving shelterbelts and mitigating wind erosion, to ultimately prevent future conditions like those seen during the 1930s.

**Author Contributions:** Conceptualization, M.W.V.B., B.C.R., and H.Z.; methodology, M.W.V.B. and B.C.R.; analysis, M.W.V.B; writing—original draft preparation, M.W.V.B; writing—review and editing, B.C.R. and H.Z.; supervision, B.C.R. and H.Z.; funding acquisition, B.C.R. and H.Z.

**Funding:** This research was supported by the NSF Experimental Program to Stimulate Competitive Research (EPSCoR) grant (#IIA-1355466), the USDA National Institute of Food and Agriculture through the Agricultural and Food Research Initiative (AFRI) Sustainable Bioenergy Program (Project No: 2013-03902, 2015-67020-23175), the AmericaView program by the U.S. Geological Survey under Grant/Cooperative Agreement No. G18AP00077, and the University of North Dakota's Department of Geography and GISc.

**Acknowledgments:** We would like to thank Paul Bjorg at the Grand Forks USDA Natural Resource Conservation Service, Steve Sagaser with the NDSU Extension Service Agency for Grand Forks County, and Greg C. Liknes at the U.S. Forest Service for their help interpreting our results. We would also like to thank P.E. Todhunter and B.J. Goodwin who served on the thesis committee, and Earl Klug for his shared interest and insight of North Dakota shelterbelts.

**Conflicts of Interest:** The authors declare no conflict of interest. The funders had no role in the design of the study; in the collection, analyses, or interpretation of data; in writing the manuscript, or in the decision to publish the results.

## Appendix A

The use of shelterbelt area as a measurement for examining shelterbelt change has a few challenges. In Section 2.4 we addressed one of these challenges by removing changes in shelterbelt area caused by increases or decreases in the tree canopy. Some studies have used linear lengths of shelterbelts as a metric for measurement [11,12], however these measurements were done in the field, or manually digitized by an individual. Using GEOBIA allowed for the automation of shelterbelt feature extraction from imagery. The GEOBIA produced polygon shelterbelts instead of lines, which were measured in terms of area [21–26]. To determine if shelterbelt area ($m^2$) can serve as a proxy of shelterbelt length (m),

we conducted a regression analysis (Figure A1). First, we measured the length of each of the shelterbelt polygons. We measured the diameter of a circle that would fit exactly around each shelterbelt polygon and considered the circles diameter as an approximate length for each shelterbelt (Table A1). We then examined the relationship between shelterbelt length and area. Our result shows that shelterbelt areas strongly correlate to the lengths of shelterbelt polygons with $R^2$ of 0.759, 0.779, 0.791 for 1962, 2014, and 2016, respectively. These data suggest that, although length is only one-dimension, shelterbelts tend to exist in long linear arrays and have less variation in their second-dimension. Therefore, we used shelterbelt area as an acceptable proxy for shelterbelt length to measure change.

**Table A1.** Statistical summary of approximated shelterbelt lengths and measured polygon areas for the three years of data.

| Year | Measured Polygon Area | | | | Approximated Length | | | |
| --- | --- | --- | --- | --- | --- | --- | --- | --- |
| | Min (m2) | Max (m2) | Mean (m2) | St. Dev. (m2) | Min (m) | Max (m) | Mean (m) | St. Dev. (m) |
| 1962 | 50.3 | 129,227.3 | 8758.6 | 10,615.9 | 11.4 | 1730.9 | 450.9 | 315.5 |
| 2014 | 50.0 | 92,130.0 | 8907.2 | 9634.6 | 10.3 | 1788.6 | 436.2 | 296.6 |
| 2016 | 50.0 | 92,130.0 | 9011.6 | 9692.2 | 10.3 | 1763.6 | 440.1 | 296.8 |

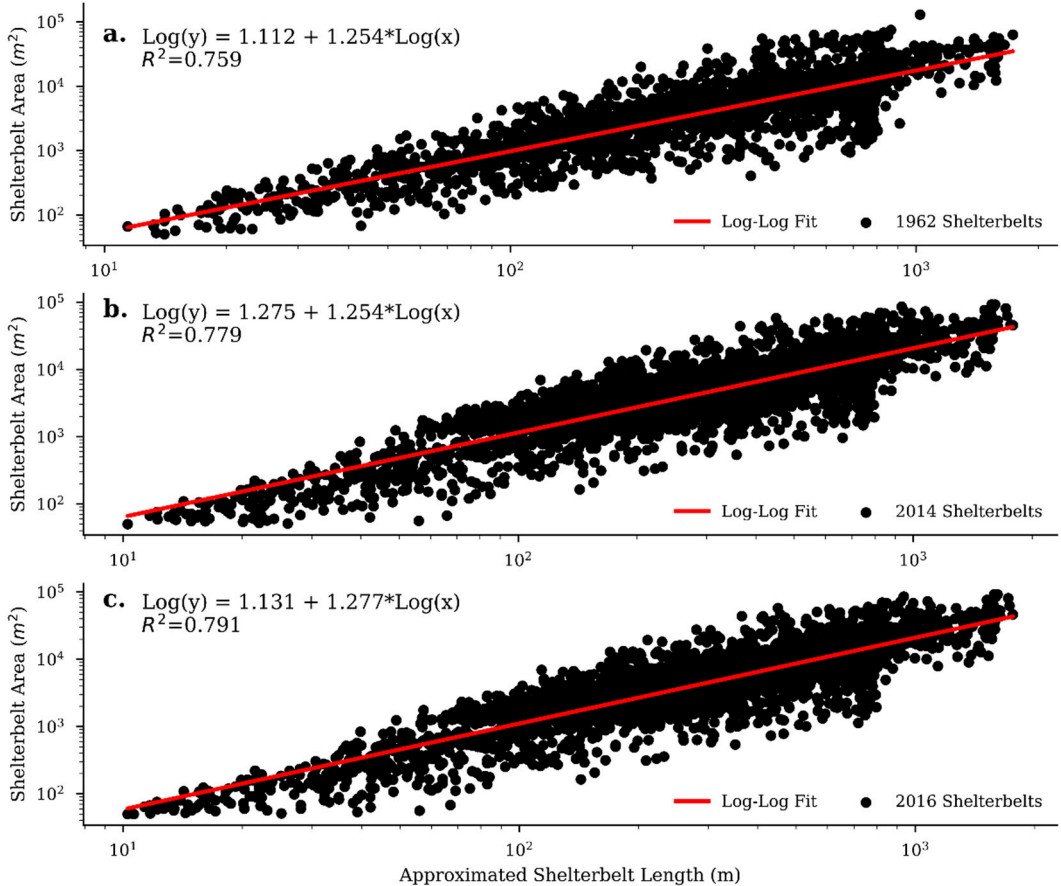

**Figure A1.** A regression analysis assessing the relationship between shelterbelt length and area. The two are highly correlated ($p < 0.0001$) across all three years: (**a**) 1962, (**b**) 2014, and (**c**) 2016, which allows area to serve as a proxy for shelterbelt length.

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
