# Peer review of "Detection of Shelterbelt Density Change Using Historic APFO and NAIP Aerial Imagery"

_remotesensing, doi:10.3390/rs11030218_

Round 1
Reviewer 1 Report
The work "Detection of recent shelterbelt density change in Grand Forks County, North Dakota, USA" is an interesting application of manual digitization and geographic object-based image analysis for the evaluation of the shelterbelt s presence and evolution. I found the work well developed and clear in all its parts.
I have a few detailed comments written directly on the text (see attached comments file).

Reviewer 2 Report
The authors investigated a very interesting topic, namely the shelterbelt density dynamics over last 52 years (1962 to 2014) but also recent changes in shelterbelt density over 2 years (2014-2016).The Ms is very well written, organized and easy to follow and read. The analyses seem to be performed properly. I have several minor comments/questions: L.158-191. It is not clear for my why the authors considered only PH and surface geology as main factors influencing the shelterbelts density. What about the climate (temperature, precipitation) or altitude range or pollution? For example if there were drought events/years, why low precipitation of the last years was not taken into account to investigate the effect on the reducing density from 2014 to 2016? In my opinion, it is also very important as the authors to add short information about the status of the shelterbelts , like species, or planting density or about the age (maybe the planting density was higher in the last period and now could be just the effect of competition among trees, or the species used for planting in the last time are more sensitive to climate change or pollution, or the trees now are old and started to die, or the new established trees were affected by frost damages in the first years, etc.). Please add the formula for ANOVA model and if and how were tested the assumptions for ANOVA (normality and homoscedasticity) and using which softare. In Discussion section I think the authors should focuss more on the explanation/factors that provoked the decreasing of shelterbets density over last 2 years.Author Response
attached

Reviewer 3 Report
Authors have used Geographic Information Systems (Spatial Analysis) and Remote Sensing (Object-based image analysis) to map and assess shelterbelts in Grand Forks County, North Dakota. This is encouraging that authors have selected one of the understudied topics (shelterbelts) in soil erosion and remote sensing field. The manuscript is well written. I suggest authors address below provided minor issues before the publication.
Comments
Please mention dominant tree species in shelterbelt in study area section.
Line 93. Is there any reason behind selecting the year 1962, 2014 and 2016 in the study?
Line 103. Please mention criteria used to define or digitized shelterbelt
Line 96, Line 105. Authors use 25 x 25 cm spatial resolution panchromatic images, and 1x 1 m spatial resolution NAIP imagery. Did authors use resampling technique to match the resolution? Which resampling method authors used?
Line 163. Please provide a full form of NRCS
Line 193. I suggest the authors include an accuracy assessment part in the method section and report the result of accuracy assessment in the result section.
Line 345. Please elaborate CP5 and CP16
